# Involvement of Kynurenine Pathway in Hepatocellular Carcinoma

**DOI:** 10.3390/cancers13205180

**Published:** 2021-10-15

**Authors:** Shivani Krishnamurthy, David Gilot, Seong Beom Ahn, Vincent Lam, Joo-Shik Shin, Gilles Jackie Guillemin, Benjamin Heng

**Affiliations:** 1Faculty of Medicine, Health and Human Sciences, Macquarie University, Sydney 2109, Australia; shivani.krishnamurthy@hdr.mq.edu.au (S.K.); charlie.ahn@mq.edu.au (S.B.A.); vincent.lam@mq.edu.au (V.L.); gilles.guillemin@mq.edu.au (G.J.G.); 2INSERM U1242, University of Rennes, 35000 Rennes, France; david.gilot@univ-rennes1.fr; 3Tissue Pathology and Diagnostic Oncology, Royal Prince Alfred Hospital, Faculty of Medicine, University of Sydney, Sydney 2006, Australia; JooShik.Shin@health.nsw.gov.au

**Keywords:** primary liver cancer, kynurenine pathway, immune evasion, indoleamine 2,3 dioxygenase 1, tryptophan 2,3 dioxygenase 2, IDO inhibitor

## Abstract

**Simple Summary:**

The kynurenine pathway (KP) is a biochemical pathway that synthesizes the vital coenzyme, nicotinamide adenine dinucleotide (NAD^+^). In cancer, the KP is significantly activated, leading to tryptophan depletion and the production of downstream metabolites, which skews the immune response towards tumour tolerance. More specifically, advanced stage cancers that readily metastasize evidence the most dysregulation in KP enzymes, providing a clear link between the KP and cancer morbidity. Consequently, this provides the rationale for an attractive new drug discovery opportunity for adjuvant therapeutics targeting KP-mediated immune tolerance, which would greatly complement current pharmacological interventions. In this review, we summarize recent developments in the roles of the KP and clinical trials examining KP inhibition in liver cancer.

**Abstract:**

As the second and third leading cancer-related death in men and the world, respectively, primary liver cancer remains a major concern to human health. Despite advances in diagnostic technology, patients with primary liver cancer are often diagnosed at an advanced stage. Treatment options for patients with advanced hepatocarcinoma (HCC) are limited to systemic treatment with multikinase inhibitors and immunotherapy. Furthermore, the 5-year survival rate for these late-stage HCC patients is approximately 12% worldwide. There is an unmet need to identify novel treatment options and/or sensitive blood-based biomarker(s) to detect this cancer at an early stage. Given that the liver harbours the largest proportion of immune cells in the human body, understanding the tumour–immune microenvironment has gained increasing attention as a potential target to treat cancer. The kynurenine pathway (KP) has been proposed to be one of the key mechanisms used by the tumour cells to escape immune surveillance for proliferation and metastasis. In an inflammatory environment such as cancer, the KP is elevated, suppressing local immune cell populations and enhancing tumour growth. In this review, we collectively describe the roles of the KP in cancer and provide information on the latest research into the KP in primary liver cancer.

## 1. Primary Liver Cancer

Primary liver cancer is the second leading cause of cancer mortality in men and the sixth most commonly occurring cancer worldwide, with an estimated 905,677 cases and 830,180 deaths in 2020 [1]. It is a tumour that develops in the liver and is known to be highly invasive and spread to other organs such as the lungs, bone marrow, lymph nodes, and brain [2,3,4]. Hepatocellular carcinoma (HCC), accounting for more than 75% of all primary liver cancer cases, and intrahepatic cholangiocarcinoma (ICC), which accounts for a lesser proportion, approximately 12–15% of all liver cancer cases, are the two main histological types of this malignancy [5]. HCC arises from hepatocytes in the liver and is the most common cause of death in people with a history of chronic liver disease [6] or cirrhosis [7].

The global burden of liver cancer-related mortality is increasing worldwide, with an estimation of >1 million diagnosed with this cancer annually by 2025 [8,9]. The highest HCC incidence and mortality rates are observed in Africa and East Asia, although a growing trend in incidence rates has been observed in western countries, including the USA and parts of Europe [10]. In Australia, the incidence rate of primary liver cancer has increased 5-fold from 2003 to 2011. According to the Australian Institute of Health and Welfare’s burden of cancer report, this cancer is a significant health threat and a burden to the Australian community [11]. A recent study showed that the age-adjusted incidence of HCC increased from 1.38/100,000 persons in 1982 to 4.96/100,000 in 2014 [12]. Incidence of HCC is up to four times higher in men compared to women and is projected to be the fifth and sixth most common cause of cancer death in Australian men and women, respectively in 2020. The gender discrepancy in primary liver cancer incidence can be attributed to biological and behavioural risk factors [13].

Important risk factors are chronic hepatitis B virus (HBV) or hepatitis C virus (HCV) infections, liver cirrhosis, chronic alcohol consumption, metabolic-associated fatty liver disease (MAFLD), and non-alcoholic steatohepatitis (NASH) [14]. HCC predominantly develops in the setting of cirrhosis and chronic liver diseases. Cirrhosis of the liver caused by any liver disease is a major risk factor, and HCC is the primary cause of death in hepatic cirrhosis patients [15]. The most common risk factor is chronic viral hepatitis [16,17,18], with HBV infection accounting for approximately 50% of the HCC cases. However, HBV vaccinations have reduced the risk associated with HBV-induced HCC [19,20]. Chronic HCV patients with cirrhosis or chronic liver damage are at higher risk of developing HCC [21]. However, a significant decrease in the risk of HCC attributed to HCV infections has been observed because of effective antiviral drugs [22]. Additionally, higher prevalence of obesity- or diabetes-related MAFLD and NASH (the most severe form of MAFLD) is also driving the increase in HCC incidence rates [23,24,25,26]. Studies suggest that older age is another important risk factor that increases the risk of developing primary liver cancer [27,28,29,30]. Statistical epidemiology shows that primary liver cancer patients mostly comprise individuals above 50 years, with mean onset age increasing from 58.2 years in 1990 to 62.5 years in 2017 [31].

### HCC Stages and Its Prognosis

Overall survival for HCC patients is poor, with a 5-year relative survival rate of 34% for patients diagnosed with localized tumour mass, 12% for patients with regional cancer that has spread outside the liver to surrounding tissues or lymph nodes, and 3% for patients diagnosed with distant or metastasized liver cancer [32]. The Barcelona Clinical Liver Cancer (BCLC) staging system is widely accepted and used to identify the stage of HCC based on tumour characteristics and burden, the Child–Pugh score of hepatic function, and patient performance status [33]. The median survival time for HCC patients can vary according to the stage of cancer diagnosed. Based on the BCLC staging system, these values are more than 6 years for early stage (0 and A), 26 to 30 months for intermediate stage (B), 12 to 19 months for advanced stage (C), and nearly 3 months for end-stage (D) HCC after receiving treatment (Figure 1) [8].

Surgical resection or partial hepatectomy [34], laparoscopic liver resection [35], and liver transplantation [36] are the most common treatments used for early stage HCC patients (i.e., BCLC stage A), when the tumour mass is more than 2 cm but less than 5 cm in size and is confined to the liver, with no evidence of vascular invasion. Radiofrequency ablation is the primary treatment of choice for single tumours less than 2 cm in size (BCLC stage 0) and is also an alternative for early stage HCC patients unsuitable for surgery or liver transplantation due to the presence of multiple tumour nodules and liver dysfunction [37].

Unfortunately, patients with HCC are often asymptomatic in the early stages; hence, detecting early stages of cancer in patients remains a challenge. A combined diagnostic approach consisting of ultrasound imaging, magnetic resonance imaging, computed tomography, and detecting alpha-fetoprotein (AFP) levels in patient sera is used to diagnose cancer and predict HCC prognosis [38]. AFP is a type of glycoprotein that is produced by embryonic endoderm tissue cells and is usually in high concentrations in maternal serum during foetal development [39,40]. This concentration of AFP drops during adulthood due to the inability of mature hepatocytes to synthesize this glycoprotein [40]. Transformed cancer cells including hepatocytes can regain this ability to synthesize AFP and have therefore been used as blood-based biomarkers for HCC diagnosis [41]. However, this biomarker is not effective in detecting patients with a low concentration of AFP (AFP <20 ng/mL), such as during early stage HCC, and a portion of advanced HCC, where AFP remains low throughout disease progression [42]. A promising alternate blood biomarker is glypican-3 (GPC-3). GPC-3 is a cell-surface proteoglycan that is highly expressed in embryonic tissues and is involved in cell proliferation and survival during foetal development [43]. In adults, GPC-3 expression is only limited to lung, ovary, mesothelium, mammary glands, and kidney [44,45]. However, high levels of GPC-3 expression are observed in HCC tissues but not in healthy adult liver, and it is a commonly used immunohistochemical marker to detect the degree of HCC tumour differentiation [46,47]. Although studies have shown 83.4% sensitivity in HCC [48], the diagnostic use of GPC-3 as an HCC biomarker remains controversial due to conflicting results [49,50,51]. A delay of as little as three months in diagnosis can result in the cancer progressing to later stages and, more importantly, it reduces patient survival rate. Focusing on early diagnosis is important to increase patients’ survival rate rather than treatment options. [52]. Other locoregional treatment strategies for some early and intermediate HCC patients (BCLC stage B) who are not fit to undergo surgery or transplantation include trans-arterial chemoembolization (TACE) [53], local radiotherapy, or a combination approach of laparoscopy with TACE or radiotherapy is used to prevent from further cancer progression [34].

Most HCC cases are diagnosed at advanced stages (BCLC stage C and D) when the tumours are too aggressive for surgical resection and have metastasized to other organ sites. Systemic treatment, which includes molecular-targeted therapy, remains a recommended treatment for locally advanced or metastatic unresectable HCC tumours [34]. To date, the first-line drug treatments for advanced HCC patients include sorafenib [54], lenvatinib [55], and atezolizumab (anti-PDL1 antibody) in combination with bevacizumab (anti-VEGF antibody) [56]. The recent IMbrave150 trial reported that patients treated with the combination regimen of atezolizumab and bevacizumab showed improved overall survival and progression-free survival compared to sorafenib. The most common treatment-related adverse events observed with combination immunotherapy are fatigue, pain, loss of appetite, and diarrhoea [57]. On the basis of these positive findings from the trial, the Therapeutic Goods Administration (TGA)-approved regimen has now been extensively used to treat patients with unresectable HCC and was added to the Australian Pharmaceutical Benefits Scheme (PBS) program in 2020 [58]. While there has been significant improvement in treatment opportunities over the last decade, this malignancy is associated with a high recurrence rate and poor overall survival. Clinical trials evaluating the efficacy and safety of immune-therapeutic drugs such as pembrolizumab or nivolumab for advanced liver cancer treatment failed to improve overall survival of patients and significant immune-related adverse side effects were observed, resulting in failure of the clinical trials [59,60].

Although the understanding of the disease and treatment opportunities for HCC have drastically improved over the last decade, this malignancy remains a fatal disease worldwide. There is an urgent need to identify a specific set of biomarkers to (1) detect early stage HCC with high accuracy in patients and (2) to effectively allow the assessment of response to treatment to rapidly estimate whether a patient responds to treatment. Identification of novel and specific diagnostic set of biomarkers to detect patients who may be at risk and with early stage HCC, prognostic predictors that can effectively distinguish between patients with favourable or unfavourable prognosis in the same tumour stage, and more specific treatment targets are all critical. An important aspect to consider is the unique relationship between the liver and the immune system. The liver is a critical immunological frontline of the body, where complex immunological activity occurs to prevent infection in the body [61,62]. Interestingly, some biochemical pathways promote tumour tolerance by decreasing the recognition of cancer antigen, inducing immune suppression and chronic inflammation. Notably, an interesting biochemical pathway that mediates tumour tolerance is the kynurenine pathway (KP) of tryptophan (TRP) metabolism. Elevation of KP activity by tumour cells suppresses the local immune response and enhances tumour survival and invasion [63,64]. This review will examine the role of KP in HCC progression. Understanding how HCC manipulates immune-suppressive KP may lead to the identification of potential therapeutic targets for HCC.

## 2. The KP

TRP is one of the eight essential amino acids that are only obtainable through the diet [65]. TRP and its metabolites play a critical role in various cellular growth and maintenance processes. Up to 90% of the TRP is catabolized by the KP to produce nicotinamide adenine dinucleotide (NAD^+^), an important enzyme co-factor involved in the regulation of important cellular processes (Figure 2) [66]. KP is tightly regulated under a healthy physiological state and produces various metabolites with immune-suppressive and redox activity. These metabolites include kynurenine (KYN), kynurenic acid (KYNA), 3-hydroxykynurenine (3-HK), anthranilic acid, 3-hydroxyanthranilic acid (3-HAA), picolinic acid, and quinolinic acid (QUIN) [67]. The pathway begins with three rate-limiting enzymes, indoleamine 2,3-dioxygenase (IDO1) [68], indoleamine 2,3 dioxygenase 2 (IDO2) [69,70], and tryptophan 2,3-dioxygenase (TDO2) [71] that catabolise the substrate TRP to KYN.

Although the three rate-limiting enzymes catabolise the same substrate, TRP, they each have different inducers and regions of expression. In normal physiological conditions, IDO1 enzyme expression is limited to endothelial cells in the lungs and placenta, epithelial cells scattered in the female genital tract and mature dendritic cells in secondary lymphoid organs, and is known to be induced by interferon-gamma (IFN-γ) [72]. Compared to IDO1, IDO2 enzyme expression is restricted and confined to hepatocytes, bile duct, neuronal cells of the cerebral cortex, and kidneys [73]. While IDO1 and IDO2 share 43% gene similarity, IDO1 remains the dominant enzyme [69]. Interestingly, the activity of IDO2 elevates when the IDO1 gene is deleted [74]. The third rate-limiting enzyme, TDO2, is primarily expressed in liver, and is the major enzyme to regulate systemic TRP levels in the liver [75,76]. TDO2 enzyme expression is known to be induced partly by glucocorticoids and its substrate TRP [77]. Though these rate-limiting enzymes are cytosolic, their enzymatic activity induces TRP metabolism and accumulation of KP metabolites in the extracellular space, which is facilitated by specific amino-acid transporters [78]. In an inflammatory environment such as cancer, KP is highly activated, resulting in depletion of local TRP in the tumour micro-environment. This process facilitates tumour cells to evade immune detection by reducing the proliferation of effector T lymphocytes and favouring the differentiation of regulatory T (T_regs_) cells [79].

### Involvement of the KP in Cancer

After the discovery that placental IDO1 was the key enzyme mediating immune suppression in maternal–foetal tolerance in 1998, the research focus was expanded to examine whether the KP was involved in immune evasion and cancer [80,81]. Indeed, the KP is frequently dysregulated in cancer and suppresses tumour surveillance in two different mechanisms. The first mechanism involves the overexpression of the rate-limiting enzymes IDO1 and TDO2 to deplete TRP within the tumour microenvironment. TRP is one of the amino acids required for the survival and proliferation of immune T-cells such as T helper (T_h_) and cytotoxic T-cells (T_c_). Therefore, immune surveillance will be strongly suppressed in a TRP-deprived tumour microenvironment driven by an overactive IDO1/TDO2 tumour [82]. A study by Uyttenhove et al. confirmed overexpression of IDO1 in various human cancer tissues and cell lines, suggesting that was involved in protecting tumours from immune detection [83]. The overexpression of IDO1 in tumours has been suggested to be induced by the IFN-γ generated by tumour-infiltrating T-cells as an adaptive resistance mechanism [84]. Syngeneic animal studies showed that treatment of the IDO1 inhibitor 1-methyltryptophan (1-MT) limited the growth of IDO1-overexpressed tumours [83,85]. A subsequent breast cancer animal model study by Muller et al. demonstrated that combined treatment with 1-MT and cancer chemotherapeutic drug paclitaxel slowed down the tumour growth progression by 30% [86]. Importantly, they observed that the efficacy of this combination therapy was highly dependent on the presence of T-cells, and the inhibition of IDO1 could potentiate the efficacy of chemotherapy.

Apart from IDO1, overexpression of TDO2 in tumour cells has been shown to facilitate immune escape. TDO2 mRNA expression was detected in different types of tumours including hepatocarcinoma [87], glioblastoma [88], breast cancer [89], and colorectal cancer [90,91]. These studies also demonstrated that TDO2 was responsible for the depletion of TRP in IDO-negative tumours to evade immune surveillance [63,88,92]. This notion was supported by an animal model study by Pilotte et al., who showed that treatment using TDO2 inhibitor in an animal model reversed the TDO2-mediated immune evasion mechanism and prevented the growth of TDO2-overexpressing tumours [92]. Consequently, this led to further studies exploring new TDO2 inhibitors for use in the treatment of TDO2-overexpressing cancer [93,94,95].

Though the role of the IDO2 enzyme in cancer remains less understood, studies have shown that IDO2 expression is upregulated in certain malignancies such as colon cancer, gastric and renal cancer [96], pancreatic cancer [97], non-small cell lung cancer [98], and may have roles in tumour immune escape, facilitating cancer cell proliferation and metastasis. Sorensen et al. described the immunogenic role of IDO2 by demonstrating the presence of spontaneous T_c_ reactivity against IDO2 in healthy and cancer patient blood samples, and reported that IDO2 supported T_regs_ cells generation that was induced by human dendritic cells [99].

The second mechanism of KP-mediated tumour evasion involves the bioactive KP metabolites KYN, 3HK, 3-HAA, and QUIN. Studies have shown that these metabolites can promote tumour proliferation and modulate the immune cell population. KYN, the first metabolite of KP, can function as an endogenous ligand to activate the aryl hydrocarbon receptor (AhR) in an autocrine/paracrine fashion, and emerging evidence points toward the tumour-promoting role of KYN-mediated activation of the AhR [100,101]. AhR is a ligand-activated transcription factor of the basic helix–loop–helix (bHLH) Per–Arnt–Sim (PAS) family [102]. It is expressed in many immune cells and plays a vital role in regulating various immune functions in a wide range of physical and pathological processes [103,104,105]. Activation of AhR may facilitate cancer cell proliferation, tissue invasion, metastasis, and angiogenesis [106]. The KYN-AhR signalling pathway can suppress the differentiation and activity of immune cells, resulting in an impaired immune response against tumours, leading to tumour immune tolerance [107]. Various studies have demonstrated the importance of KYN-AhR activation in IDO1- or TDO2-expressing tumour cells and its role in enhancing cancer cell survival and motility. These studies suggested that TDO2-expressing cancer cells escape immune surveillance by activating AhR in various immune cells including dendritic cells, macrophages, natural killer cells, innate lymphoid cells, T_c_ cells, and T_regs_ cells [108,109]. Opitz et al., found that murine tumours in AhR-proficient mice expressing high AhR and TDO2 expression levels had an enhanced tumour growth rate by suppressing the infiltration of antitumour immune cells, increasing levels of inflammatory cytokines. Furthermore, the study suggested that the TDO2-Kyn-AhR signalling pathway might also be involved in other malignancies, including sarcoma, bladder cancer, cervix cancer, colorectal cancer, lung, and ovarian cancer [88]. Moreover, Ulrike et al. revealed that IDO1 enzyme expression was induced by inflammatory cytokines such as Interleukin 6 (IL-6), and could activate an autocrine-positive inflammatory feedback loop (IDO-AhR-IL-6-STAT3 signalling pathway) that could promote tumour growth and survival [110].

In addition to KYN, kynurenic acid (KYNA) is also an endogenous AhR ligand [111]. In the presence of IL-1β, KYNA binds to AhR and induces production of IL-6, which may also contribute to the IDO-AhR-IL-6-STAT3 autocrine-positive inflammatory feedback loop mentioned earlier. Interestingly, the production of KYNA may not be limited to just via KP but rather through an alternate TRP metabolism mediated by Interleukin-4-induced gene 1 (IL4I1) in a cancer setting. Sadik et al. revealed that IL4I1 was elevated in cancers such as melanoma. An IL4I1-driven AhR activity though KYNA increases tumour cell motility and T-cell proliferation [112]. Given that the activity of IL4I1 is independent of the KP and can limit antitumor immune cell response [113], inhibiting the formation of KYNA metabolite either via the KP or through IL4I1 gene reaction may be necessary to block the activation of AhR in cancer.

The KP metabolites downstream of KYN, including 3-HK, 3-HAA, and QUIN have been shown to inhibit T-cell proliferation and activation. A study by Fallarino et al. showed that 3-3-HAA and QUIN could induce selective apoptosis in T_h_1 cells and thymocytes of effector T-cell population in vitro by the activation of caspase-8 activity and the release of cytochrome c from mitochondria [114]. The 3-HAA also significantly inhibits CD8^+^ T-cell proliferation stimulated through cytokines by driving the T-cells to a proliferative arrest and directly inhibiting the phosphorylation of phosphoinositide-dependent kinase 1 and preventing the activation of nuclear factors after T-cell receptor stimulation [115]. A study by Favre et al. showed that 3HAA also disturbed the balance between T_h_ and T_reg_ cell populations, driving them towards an immunosuppressive T_reg_ pathway in vitro [116]. Furthermore, a later study by Zaher et al. confirmed that 3HK and 3HAA suppressed CD4^+^ T-cell proliferation along with significant T-cell death [117].

## 3. Involvement of the KP in Chronic Liver Disease and HCC

The role of KP in liver diseases has been gaining interest in the recent years. A number of studies have measured high KP activity in chronic liver diseases such as primary biliary cirrhosis, HCV-associated chronic hepatitis, and liver cirrhosis [118,119]. Claria et al. [120] reported that KP activity was elevated in patients with acute decompensation and acute-on-chronic liver failure, and was associated with pathogenesis and mortality in cirrhotic patients. The study concluded that elevated KP activity may be used as an independent prognostic predictor of poor clinical outcomes in cirrhotic patients. In contrast, elevated IDO1 activity during early stages of the HBV infection in hepatocytes was reported to significantly reduce viral replication and enhance the protective immune response [121].

Although the liver is a site of robust immunological activity, liver cancer cells can remain undetected and proliferate. This suggests that these cancer cells can evade local immune surveillance, possibly by using the KP, as observed in various malignancies. Although the research on KP and HCC is limited, the activity of the three upstream enzymes of the pathway, including IDO1, TDO2, and KMO enzymes, has been extensively studied in HCC cells and tissue specimens. These study findings revealed that IDO1, TDO2, and KMO enzyme activity was upregulated in HCC (Table 1).

### 3.1. IDO1

The immunological and prognostic roles of IDO1 in HCC were first investigated by Ishio et al. in 2004 [122]. The results showed that IDO1 mRNA expression was strongly induced in tumour-infiltrating cells of the HCC tumour, which might facilitate an antitumour immune reaction and the expression of IDO in tissue specimens of HCC patients significantly correlated with better recurrence-free survival rates. A later study by Ke Pan et al. observed elevated IDO1 enzyme mRNA and protein expressions in liver tumour and its adjacent normal tissues compared to distant non-involved normal tissues, suggesting that IDO1 overexpression was confined to the tumour microenvironment [123]. A potential explanation for the confined IDO1 expression could be due to the presence of inflammatory cytokine(s) in the tumour microenvironment that activate IDO1 activity. Indeed, a later study by Li et al. demonstrated that IDO1 enzyme expression was observed only in IFN-γ-stimulated HCC cells through the IFN-γ-JAK2-STAT1-signalling pathway. Moreover, high IDO1 expression in HCC positively correlated with abundance of CD8+ T-cells, thus reflecting an antitumour immune response and suggesting that IDO1 could be used as a favourable prognostic indicator for HCC patients [124]. Lastly, Brown et al. suggested that IDO1 enzyme inhibitors in combination with immune checkpoint inhibitors could be a novel treatment approach for liver cancer treatment [125].

### 3.2. TDO2

A recent study conducted by Hoffman et al., showed that the majority of the tumour cells in HCC tissues expressed TDO2 in HCC [126]. This study demonstrated the immune-regulatory role of the TDO2 enzyme in HCC tumour cells, and suggested that the TDO2 enzyme was a promising immunotherapy treatment target for HCC. Another study by Li et al. characterized the overexpression of TDO2 enzyme in HCC cancer cells and suggested that it might play a vital role in promoting HCC cancer cell growth, migration, and invasion in vitro and in vivo [127]. Additionally, TDO2 expression was correlated with the development of the tumour, such as size, tumour differentiation, and vascular invasion. Based on these strong correlation data, the authors suggested that TDO2 expression could be used as an effective biomarker to predict overall or disease-free survival of HCC patients. Activation of AhR is associated with the loss of cell contact inhibition and changes to the extracellular matrix, and extensive studies have demonstrated that this activation induces epithelial to mesenchymal transition (EMT) in various cancers [130,131,132]. Overexpression of AhR in HCC has been shown to be associated with its tumour proliferation and invasion [133,134]. A recent study by Lei Li et al. showed that upregulated expression of the TDO2 enzyme promotes the migration and invasion capabilities of HCC cells by the KYN-AhR-mediated induction of epithelial to mesenchymal transition, a process that is vital for cancer metastasis [87].

### 3.3. KYN Levels in Patient Sera

A recent retrospective study on a cohort of HCC patients with chronic HCV infection revealed that KYN levels were elevated in HCV-mediated HCC patient sera in comparison to healthy controls (non-HCC patients). Bekki et al. observed that KYN production gradually increased when chronic HCV progressed to HCC, and suggested the potential of using serum KYN levels as a biomarker for predicting survival and prognosis in early stage HCV-mediated HCC patients [128].

### 3.4. KMO

Kynurenine 3-monooxygenase (KMO) is the immediate KP enzyme after the rate-limiting step, and it is widely distributed in the peripheral tissues of the liver and kidney, astrocytes and microglial cells situated in the brain, central nervous system [135,136], and phagocytes, including macrophages and monocytes [137]. KMO localizes to the outer membrane of mitochondria and catabolizes KYN to 3-HK. The role of KMO enzyme expression in cancer has rarely been studied in comparison to IDO and TDO2 enzymes. Liu et al. identified the oncogenic role of KMO in triple-negative breast cancer progression [138]. Moreover, high surface expression of KMO was detected in cytosol and on the cell membranes of breast cancer tissue specimens, indicating its potential as a treatment target for TNBC [139]. A recent study investigated the correlation between upregulated KMO activity and poor clinical outcomes in colorectal cancer (CRC) patients and demonstrated that KMO inhibition suppressed CRC cell proliferation in vitro [140]. On analysing KMO enzyme expression in 120 matched HCC tissue samples, Jin et al. showed that the expression of the KMO enzyme is significantly elevated in HCC tumour tissue compared to adjacent normal liver tissue. High KMO expression correlated with poor patient outcomes, which indicates that the KMO enzyme may be a significant prognostic marker in HCC patients [129]. Results from the in vitro experiment comparing KMO enzyme levels in human normal liver cells and HCC cell lines showed that KMO enzyme was upregulated in HCC cells and might play a role in promoting tumour proliferation, metastasis, and invasion. The study also demonstrated that KMO knockdown in HCC cell lines by small interfering RNA (siRNA) transfection decreased cancer cell proliferation, thus suggesting that KMO could be a novel target for HCC treatment.

### 3.5. Clinical Trials: IDO1 Inhibitors as HCC Treatment

IDO1 inhibitors are small molecule drugs that competitively block the activity of the IDO1 enzyme without inhibiting IDO2 or TDO2 [141]; several of these drugs are in clinical development. The safety and efficacy of many IDO1 inhibitors, including Indoximod, Epacadostat, Navoximod, BMS-986205, and others, have been tested in combination with other immunotherapy drugs such as pembrolizumab and nivolumab for the treatment of various metastatic cancers. Currently, two small molecule IDO1 inhibitors, BMS-986205/NCT03695250 and INCB024360 (Epacadostat)/NCT02178722, are in phase I/II clinical trial to evaluate their safety and efficacy in HCC patients [142,143]. The clinical trial NCT03695250 is a single-group assignment that examines the safety, tolerability, and efficacy of BMS-986205 with nivolumab in unresectable/metastatic HCC. It is still active but not recruiting patients; hence, the results have not been published yet. The expected treatment-related adverse events of BMS-986205 would be at grade 1–2 such as fatigue and nausea, as reported in the other trials examining the efficacy of BMS-986205 in cancer patients. Clinical trial NCT02178722 evaluated the safety, tolerability, and efficacy of Epacadostat in combination with pembrolizumab. This trial concluded that the combination regime has an acceptable safety profile in patients with advanced cancers, achieving an objective response rate in 12 of 22 cancer patients [144,145]. Treatment-related adverse events observed in 84% of the patients enrolled were of grade 1–2. The most common events were fatigue, rash, arthralgia pruritus, and nausea. This result supports additional phase 3 studies in other malignancies but not in HCC.

## 4. Conclusions

HCC is one of the few malignancies for which the risk factors have been well-established. Although patients with early stage HCC have the best median survival time and can usually be cured by resection, liver transplant, or ablation, they are often asymptomatic. Hence, most patients present with late-stage HCC and have a poor prognosis. The approved first-line treatment of late-stage HCC is multikinase inhibitors such as sorafenib, which confers a slightly longer survival time. However, this treatment is associated with substantial side effects that have a negative impact on quality of life. This therefore changes the treatment focus by combining current antitumoral drugs with immunotherapy, and this approach has significantly benefited HCC patients. A recently concluded trial examining combination therapy of atezolizumab with bevacizumab showed a significant improvement in overall survival and progression-free survival as compared to sorafenib. Since this study, it has been adopted as the first-line treatment for late-stage HCC. Considering the strong evidence of its ability to mediate immune suppression, the KP might be an alternative immunotherapy target and play a role in the progression of liver cancer, as summarized in Figure 3.

This notion is supported by clinical studies that showed an elevated KP enzyme profile in HCC cells and tumour tissue specimens, with elevated expressions associated with disease aggressiveness. Although current IDO1 inhibitor clinical trials are still in phase I/II evaluation, it is possible to suggest that the use of KP inhibitors in combination regimens may improve the survival mark of early and advanced HCC.

## Figures and Tables

**Figure 1 cancers-13-05180-f001:**
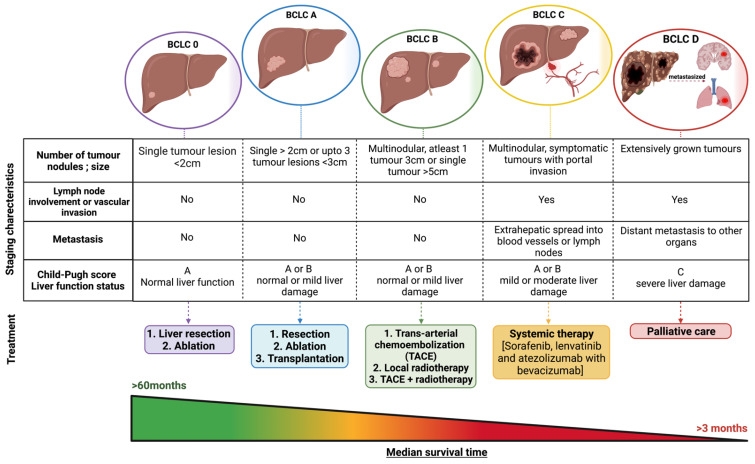
Classification of HCC and its characteristics: Based on the BCLC staging system, HCC can be classified as stages 0, A, B, C, and D. Stage A has the highest median survival time of more than 60 months while stage D has less than 4 months. Localised surgery and radiotherapy are the choice of treatments for stage 0 to B, while systemic treatment with palliative care is usually recommended for stages C to D.

**Figure 2 cancers-13-05180-f002:**
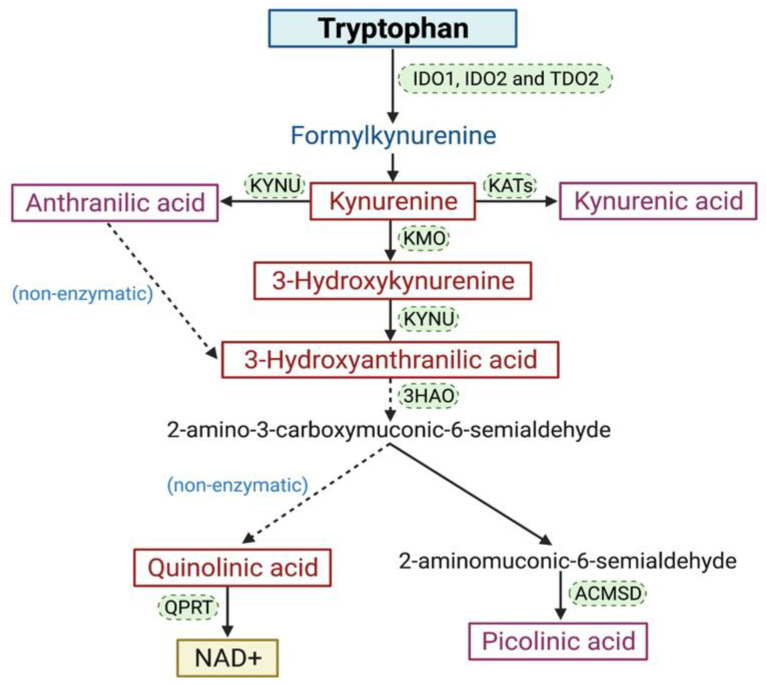
A simplified diagram of the KP: majority of TRP is catabolized through the KP to synthesize the vital energy cofactor, NAD^+^.

**Figure 3 cancers-13-05180-f003:**
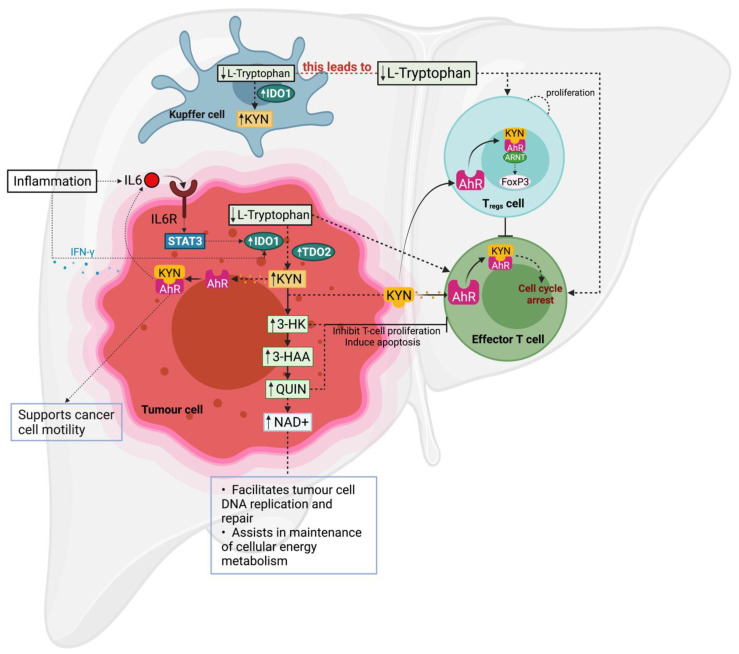
The KP-mediated immune tolerance and cancer invasion: KP promotes immune tolerance by two different mechanisms. Firstly, elevated IDO1/TDO2 enzyme activity in either tumour or immune cell depletes TRP concentration in its local tumour environment. A TRP-stripped environment induces cell arrest in T-cells while inducing differentiation and proliferation of T_reg_ cells. Secondly, downstream KP metabolites induce cell arrest in T-cells and T_reg_ proliferation by either interaction with AhR or by direct interaction with immune cells themselves. In addition to KP-mediated immune tolerance, elevated KP promotes cancer cell motility and proliferation by either overproduction of NAD^+^ for cellular repair or byactivation of AhR.

**Table 1 cancers-13-05180-t001:** Summary of all KP research carried out on HCC.

Author (Year) [Ref]	Sample Type	Sample Size	Enzyme/Metabolite Studied	Technique Used	Finding
Ishio et al. (2004) [122]	Cell lines HCC tumour specimens	4 HCC cell lines 21 HCC	IDO1	RT-PCR IHC	IDO1 expression may play a role in antitumour immune response.
Pan et al. (2008) [123]	Cell lines Tumour and distant normal liver tissue	6 HCC and human normal hepatocytes 138 HCC	IDO1	RT-PCR IHC	HCC cancer cells and surrounding noncancerous tissue express high IDO1 expression. High IDO1 expression is confined to the tumour creating an immune suppressive microenvironment.
Li et al. (2018) [124]	Cell lines Tumour tissue	2 HCC cell lines 112 HCC (HBV) *	IDO1	RT-PCR WB IHC and IF	IDO1 is expressed in HCC cells on stimulation by IFN-γ via the JAK2-STAT1 signalling pathway. High IDO1 expression indicates antitumour immune response. IDO1 is a favourable prognostic indicator.
Brown et al. (2018) [125]	Cell lines	2 HCC cell lines	IDO1	RT-PCR	IDO1 inhibitors in combination with immune checkpoint inhibitors might be an effective treatment option for HCC patients.
Hoffman et al. (2020) [126]	Cell lines Tumour and normal tissue	1 HCC cell line 171 tissue specimens	TDO2	RT-PCR WB, HPLC IHC and IF	High TDO2 expression observed in HCC tumour cells. TDO2 may be a novel immunotherapeutic target for HCC.
Li et al. (2020) [127]	Cell lines Paired tumour and adjacent normal tissues	5 HCC cell lines and 1 normal liver cell line 93 HCC	TDO2	RT-PCR WB RT-PCR WB, IHC	TDO2 is overexpressed in HCC and may be facilitating HCC progression and invasion. TDO2 enzyme can be a novel prognostic biomarker for HCC patients.
Lei et al. (2021) [87]	Cell lines Paired tumour and adjacent normal tissue	6 HCC cell lines and 1 normal liver cell 23 HCC	TDO2	RT-PCR WB Knockdown using shRNAs HPLC IHC and IF	TDO2 supports EMT of HCC cells via the KYN-AhR pathway, facilitating HCC metastasis and invasion.
Bekki et al. (2020) [128]	Serum	604 HCC * (HCV) 288 Control **	KYN	ELISA	A high level of serum KYN correlated with poor prognosis of HCC.
Jin et al. (2015) [129]	Tumour and adjacent noncancerous liver tissue Cell lines	120 matched HCC and adjacent tissue 205 HCC 5 HCC and 2 human normal liver cells	KMO	IHC RT-PCR WB Knockdown using siRNAs	High KMO expression correlated with HCC tumour aggression, recurrence, and shorter survival rate. KMO knockdown suppressed HCC progression in vitro. KMO overexpression enhanced HCC cell proliferation, migration, and invasion.

* HCC patients with chronic hepatitis B (HBV) or hepatitis C (HCV) virus infection. ** Patients with chronic hepatitis C virus infection without HCC. RT-PCR: reverse transcription-polymerase chain reaction, WB: Western blot, IHC: immunohistochemistry, IF: immunofluorescence, HPLC: high-performance liquid chromatography, shRNA: short hairpin or small hairpin RNA, siRNA: small interfering RNA.

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
