# Peer review of "Involvement of Kynurenine Pathway in Hepatocellular Carcinoma"

_cancers, 2021, doi:10.3390/cancers13205180_

Round 1
Reviewer 1 Report
The manuscript entitled “Involvement of kynurenine pathway in primary liver cancer” by Krishnamurthy S et al. described the roles of the kynurenine pathway (PK) in primary liver cancer. The authored first introduced primary liver cancer, then review the KP and its role in cancer, finally, review the role the KP in HCC. Overall, the manuscript is well written and well organized. Authors made an intense review of the articles relative to the topics. However, there are some major problems and minor issues in the manuscript.
Major comments:
1). P5, paragraph 2, line 144-145: the authors stated that sorafenib and Lenvatinib are the only available first-line drug therapies for the late staged patients. However, this is not true. FDA has already approved atezolizumab and bevacizumab as an initial treatment for people with liver cancer that has spread or that can’t be treated with surgery on May 29, 2020, and since then this combination treatment becomes the new reference standard in first-line HCC treatment.
2). The references in table 1 doesn’t match the references in the reference list. Also since the title of table 1 is summary of all KP research carried out on HCC, it should include all the references mentioned in your manuscript. Please update table 1.
Minor comments:
1). Page 1, line 13: primary liver cancer is the third leading cancer related death in world. This is not matching the statement in page 3, line 68. Please correct it in the abstract.
2). Page 2, line 42, HBC should be HCV.
3). Page 2, line 63, Tregs should be Tregs to matching page 6, line 208. Similar problem in the figure 3. Please correct Treg cell to Tregs cell. The abbreviation should be consistent through the whole manuscript.
4). The format in page 4, line 133-135 is different from other part in this manuscript.
5). In figure 2, please clarify what the dash line and solid line mean.
6). There is a typo error in page 8, line 283: “IL411” should be “IL4I1”.
7). Page 10, line 360, the research conducted by Shigemune et al should be cited as Bekki S et al since Bekki is the last name.
8). The author name Haojin as listed in page 11, line 379 should be Haojie Jin.
Author Response
We like to thank the reviewer for your helpful comments.
Our response is in the latest uploaded version in the "Response to Reviewer 1 word document" and cancers-1401716 with tracking" PDF.

Reviewer 2 Report
KP as it relates to cancer is an interesting topic.
Some comments:
- Is this review about primary liver cancer or is it about hepatocellular carcinoma? The topic appears to be the latter. There is mention if ICC (cholangiocarcinoma), but not a discussion of it. There is also no discussion of roles of KP in cholangiocytes or tumours of that cell type. Thus, the title should be adjusted to hepatocellular carcinoma or an explanation of how KP acts in cholangiocytes and in cholangiocarcinoma added.
- Abstract: More than 75%, probably 80-90%, of primary liver cancer is HCC. The 5-year survival for primary liver cancer is >19% in Australia [eg see latest data from Cancer Council NSW]. Treatment of advanced primary liver cancer is not limited to palliative; a common treatment now is atezolizumab in combination with bevacizumab in Australia, since that therapy was TGA approved and then PBS listed in 2020.
- Parts of the clinical picture are outdated. Eg page 3 uses ref 25 from 2008 for survival data. This was before sorafenib was widely used. Also, the section on disease prevalence must have international data added for this international journal. The talk on HCC therapy by A/Pr Simone Strasser at the recent GESA conference in Sydney contained up to date information; this review should be as up to date.
- Page 3 para on chronic liver diseases is a jumble. It needs to be teased out into distinct statements on HBV , on HCV, on NAFLD [now often called MAFLD] and the severe form of MAFLD called NASH. Eg there is no vaccine for HCV, but the current wording could readily be misinterpreted as implying that there is.
- Fig 1 fonts are too small. Is this an original fig or derived from a publication? The legend needs more explanation, such as stating that the ‘systemic therapy’ is the variety of therapies that include 3 multikinase inhibitors, and also atezolizumab in combination with bevacizumab. Also, explain what TACE is.
- Page 4 has a good explanation of AFP, so glycipan 3 should also be explained as a HCC biomarker.
- Page 10 line 335-7; why make this suggestion about ICI, rather than an effective therapy, after explaining on page 5 that ICI are ineffective in HCC?
- Page 11 Conclusion: Needs reconsidering and rewriting. Line 409 doesn’t make sense. The sentence in lines 412-3 is unclear, and puzzling, when considering page 5. The most exciting change in HCC therapy since sorafenib is the new therapy of atezolizumab in combination with bevacizumab and this must be included in the concluding remarks.
- Side effects of current and emerging liver cancer therapies, including Side effects or potential Side effects of manipulating the KP, should be added to this review.
- IL4I1 is sometimes misspelled as IL411 [pages 2 and 6].
- Page 4 line 71 delete ‘and others’
- Page 4 line 73 add comma after 15%
- Page 4 line 77 delete ‘been’
- Page 5 line 159 rewrite ‘a patients respond’
- Page 5 line 164 suggest change to ‘critical immunological frontline’
- Page 5 uses the word ‘factor’, which is a vague term, 3 times; consider alternate wording that provides clarity .
- Page 6 line 212 delete ‘studies have shown that’. This phrase is always redundant.
- Page 8 line 297 add to heading the words ‘chronic liver disease and’
- Page 11 line 389-391 rewrite this sentence because it is clunky and has errors of English.
Author Response
We like to thank the reviewer for your helpful comments.
Our response is in the latest uploaded version in the "Response to Reviewer 2 word document" and cancers-1401716 with tracking" PDF.

Round 2
Reviewer 1 Report
The authors have properly addressed all comments and made proper changes in the manuscript.